# A Review on Catalytic Depolymerization of Lignin towards High-Value Chemicals: Solvent and Catalyst

Yannan Wang [1,2], Lianghuan Wei [1], Qidong Hou [2], Zhixin Mo [1], Xujun Liu [3] and Weizun Li [2,*]

1   Xinjiang Biomass Solid Waste Resources Technology and Engineering Center, College of Chemistry and Environmental Science, Kashi University, Kashi 844006, China; wangyannan@nankai.edu.cn (Y.W.)
2   National & Local Joint Engineering Research Center on Biomass Resource Utilization, College of Environmental Science and Engineering, Nankai University, Tianjin 300071, China
3   State Key Laboratory of Molecular Engineering of Polymers, Fudan University, Shanghai 200433, China
*   Correspondence: liweizun@nankai.edu.cn

**Abstract:** Lignin is a type of natural aromatic material with potential application prospects obtained from lignocellulosic biomass. Recently, the valorization of lignin has received increasing attention from both industry and academia. However, there is still a challenge in the efficient valorization of lignin due to the complexity and stability of the lignin structure. Recent work has been focused on the catalytic depolymerization of lignin to explore a promising and efficient way to valorize lignin into chemicals with high value and biofuels. Considerable research has focused on catalysts, solvents, and reaction parameters during the lignin depolymerization process, which significantly affects product distribution and productivity. Thus, in a catalytic depolymerization process, both catalysts and solvents have a significant influence on the depolymerization effect. This review article assesses the current status of the catalytic hydrogenolysis of lignin, mainly focusing on the solvents and catalysts during the reaction. First, various solvents applied in the lignin depolymerization reactions are extensively overviewed. Second, the recent progress of metal catalysts as well as their supports is summarized. Furthermore, a discussion of the challenges and prospects in this area is included.

**Keywords:** lignin; catalytic depolymerization; solvent; catalyst

## 1. Introduction

In recent years, a growing need for energy and the depletion of fossil fuels has led to an energy crisis all over the world. Lignocellulosic is a promising renewable resource and has great potential in decreasing global carbon emissions [1]. Among the three major components of lignocellulosic, lignin gives plants their structural integrity and fortification. Globally, about 170 billion tons of lignin is produced annually [2], which is an unwanted constituent in the pulp and paper industry and traditionally is treated with the direct combustion and discharge method. As the world's most abundant natural aromatic material, lignin has great potential to obtain chemicals and fuels through various processes. Previous studies, however, mostly focused on cellulose- or hemicellulose-based materials as raw materials to produce value-added products [3]. There is still a long way to go before lignin can be exploited on an industrial scale due to the complicated three-dimensional lignin network and high bond energy [4]. Hence, it is necessary to develop lignin-processing methods for high-value-added chemicals.

Lignin is an intricate macromolecule mainly composed of three aromatic types of units: 4-hydroxyphenyl (H), guaiacol (4-hydroxy-3-methoxyphenyl, G), and syringyl (4-hydroxy-3,5-dimethoxy phenyl, S) [5]. The valorization of lignin mainly focuses on the production of lignin-derived phenolics due to the phenolic nature of lignin. Phenol is an essential chemical for the polymer industry, and benefits from its broad application include epoxy, polycarbonate, Bakelite, etc. [6]. There are two main directions of lignin recycling strategies. The first application is using modified lignin as function materials, such as phenolic or

epoxy resins, copolymer materials, polyurethane foams, adhesives, and carbon fibers [7]. The other strategy is transformation into value-added products and low-molecular-weight chemicals through lignin depolymerization (LD), for instance, bio-fuel, benzene–toluene–xylene, phenols, vanillin, and DMSO [8] (Figure 1a).

Over the past few decades, the series of studies on lignin depolymerization processes (LDP) has grown rapidly, including noncatalytic, catalytic pyrolysis, enzymolysis, catalytic depolymerization, and oxidation [9–11]. Although traditional processes could depolymerize lignin effectively, they require relatively high energy input, such as higher temperature or pressures with external hydrogen or oxygen. In addition, the organic solvents applied in those processes are always environmentally unfriendly. Owing to its relative inexpensiveness and high efficiency for breaking linkages in lignin (e.g., C–O–C and C–C), catalytic chemical conversion with the assistance of various catalysts is becoming a very competitive method for the generation of bio-energy and valuable chemicals [12].

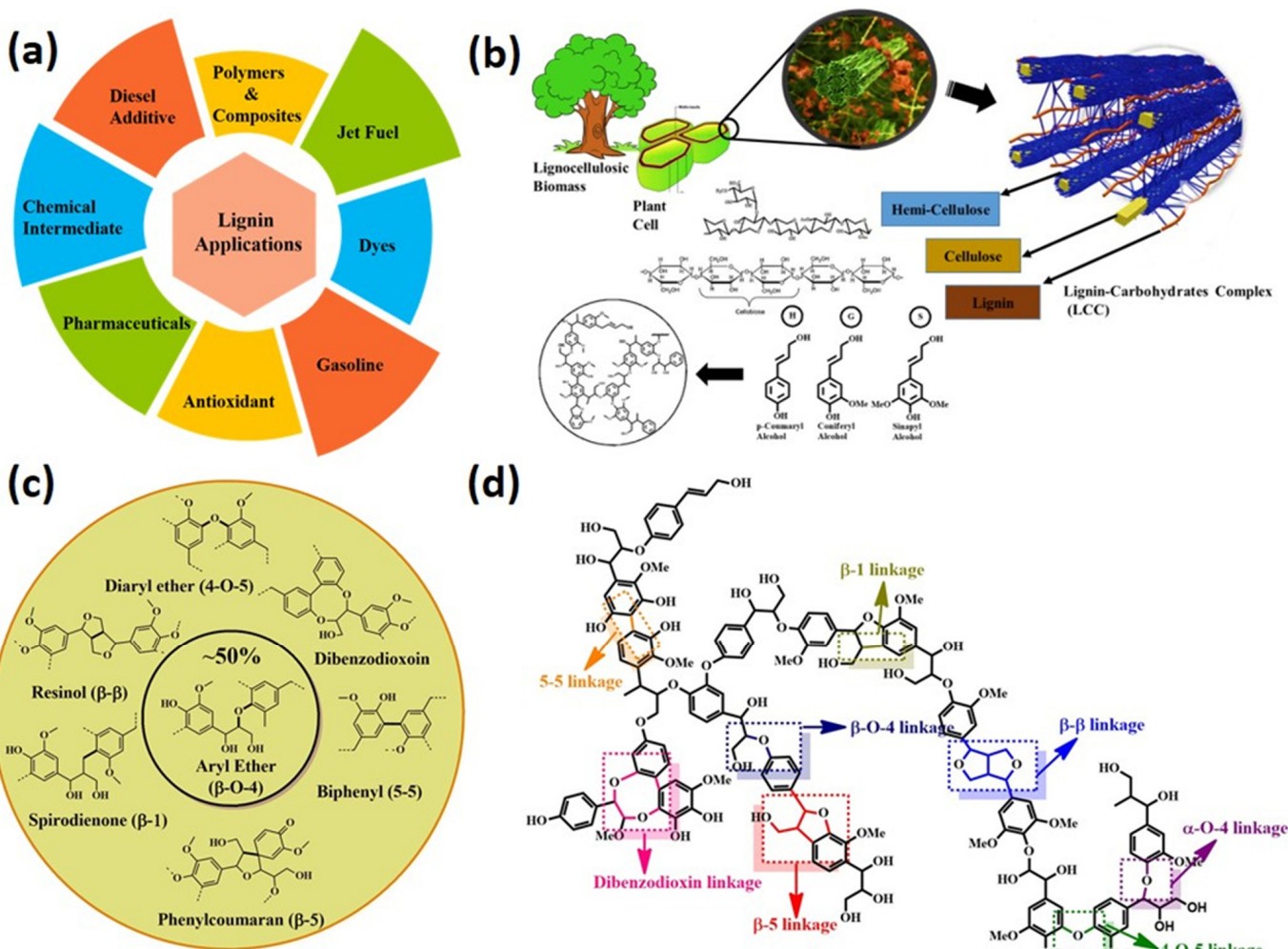

**Figure 1.** (**a**) Some materials can be produced from lignin. (**b**) The structure of lignocellulosic biomass. (**c**) Linkages between lignin molecules. (**d**) Common structure of lignin [13].

The advantage of chemical depolymerization strategies over other methods is their exceptional selectivity and conversion efficiency. According to the reaction pathway, strategies for lignin catalytic conversion could be categorized as reductive and oxidative depolymerization. LD by oxidation is decomposed by various oxidizing agents (e.g., oxygen, nitrobenzene, metallic oxide, hydrogen peroxide, etc.) to produce phenolic derivatives [14]. Electron transfer and extraction of hydrogen atoms from lignin usually occur in the reaction, which leads to a series of subsequent reactions, such as phenol oxidation, benzylic

oxidation, hydroxylation of aromatic rings, ring-opening reactions, and demethylation [15]. Oxidative depolymerization of lignin produces aromatic aldehydes (e.g., vanillin and syringaldehyde) and their acids (e.g., vanillic acid and syringic acid) [14]. Redox catalysts are employed in the process of reductively depolymerizing lignin. Along with this reaction, hydrogen or H-donor solvents are always involved, such as formic acid and alcohols. Reductive degradation is usually associated with the break of ether bonds, and as a result of degradation, phenolic compounds and lignin oligomers are primarily formed [16].

Recycling biomass into value-added materials is necessary for the circular economy. A lignin-based circular economy is shown in Figure 2. When examining the value-added chemicals from lignin, the most advantageous chemicals are benzene, toluene, xylene (BTX) [17], vanillin, and phenols. Lignin is currently valued at less than USD 50 per dry ton on the market. With a market size of approximately 300 M ton/year, the current price for BTX is in the range of USD 1000–1200/t. Similar to BTX, the value of phenols per ton is approximately USD 1500, with production estimated to be 8 million tons per year. Moreover, there is an order of magnitude higher profit potential for vanillin production from lignin than for BTX [18].

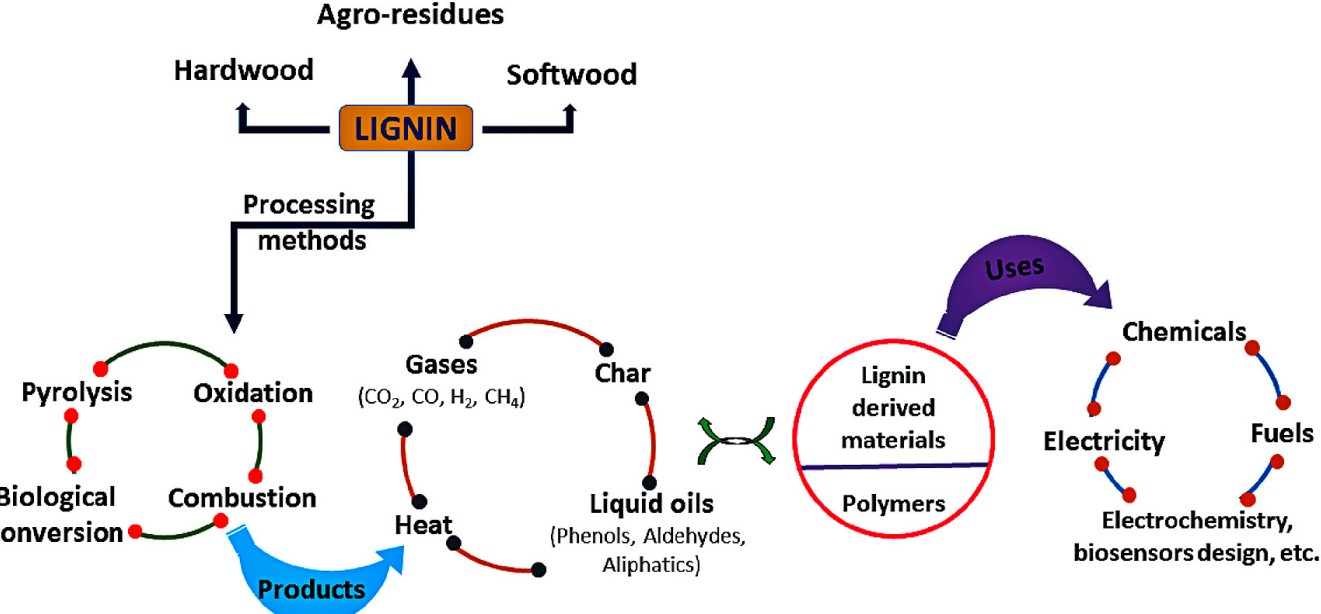

**Figure 2.** Lignin-based circular economy [17].

Hence, lignin has been converted catalytically by a variety of reaction systems in numerous studies. The mechanisms of different reactions involving solvents and catalysts are different. The performance of different solvents and catalysts can lead to different product yields, distributions, stability, and selectivity. Therefore, the purpose of this review will focus on various solvents and catalysts applied in lignin conversion. The performance of solvents and catalysts is summarized to provide a useful guide for the selection of solvents and catalysts.

## 2. Lignin Structure and Properties

As previously mentioned, there are three main units in lignin: guaiacyl (G), sytingyl (S), and p-hydroxyphenyl (H). Several C–O and C–C bonds connect them together to form hard macromolecular structures with complex crosslinks (Figure 1c,d). Because of these aromatic units, lignin has become nature's only renewable aromatic resource. The ratio of the three units varies in different plants. For instance, most softwood lignin essentially consists of G units and a lower content of H units, while lignin from grass and straw

consists of different ratios of H, G, and S units [19], and hardwood lignin contains G and S units [20].

In plant cell walls, lignin plays an important role in the structural integrity and stability of the entire plant. Lignin with a complex and rigid structure is more challenging to break down than cellulose or hemicellulose. Linkages within lignin are mainly C–O (β–O–4, α–O–4, and 4–O–5) and C–C (β–β, β–1, β–5, and 5–5) [19,21]. As C–C bonds have a higher bond dissociation energy, their cleavage is more difficult than C–O bonds [20]. The cleavage of interunit linkages is essential for the efficient utilization of lignin. Among these linkages, the β-O-4 linkage has the highest amount of native lignin [22], and most studies have been focused on its degradation [23].

The extraction of lignin from native biomass is a critical and difficult technology due to rigid and complex linkages among lignin and other constituents. Generally, different technologies obtained different types of lignin with varying degrees of structural changes, such as Kraft, alkaline, organsolv lignin or hydrolysis lignin. Figure 3 shows different extraction methods and the structure of obtained lignin [24]. Kraft lignin (KL) is a byproduct obtained from the pulp and paper mill industry, and hence, KL is the most abundant type of lignin compared with other types [25]. Generally, sulfur functionality is enriched in KL, which had a negative impact on the utilization of lignin [26]. Organosolv lignin(OL) is obtained from organosolv pulping with the assistance of organic solvents (e.g., alcohol, acetone, and ethylene) [27]. OL could be soluble in most polar organic solutions, as well as in basic solvents [8]. Soda lignin (SL) is produced as a co-product from sodium bicarbonate pulping, which is more appealing and more likely to be used as feedstock than KL [25]. Steam-explosion lignin (SEL) and pyrolysis lignin (PyL) are extracted from lignocellulose submitted to a steam explosion process and a pyrolytic process, respectively [8]. Several types of lignin and their extraction processes, as well as structural features, are summarized in Table 1 [18].

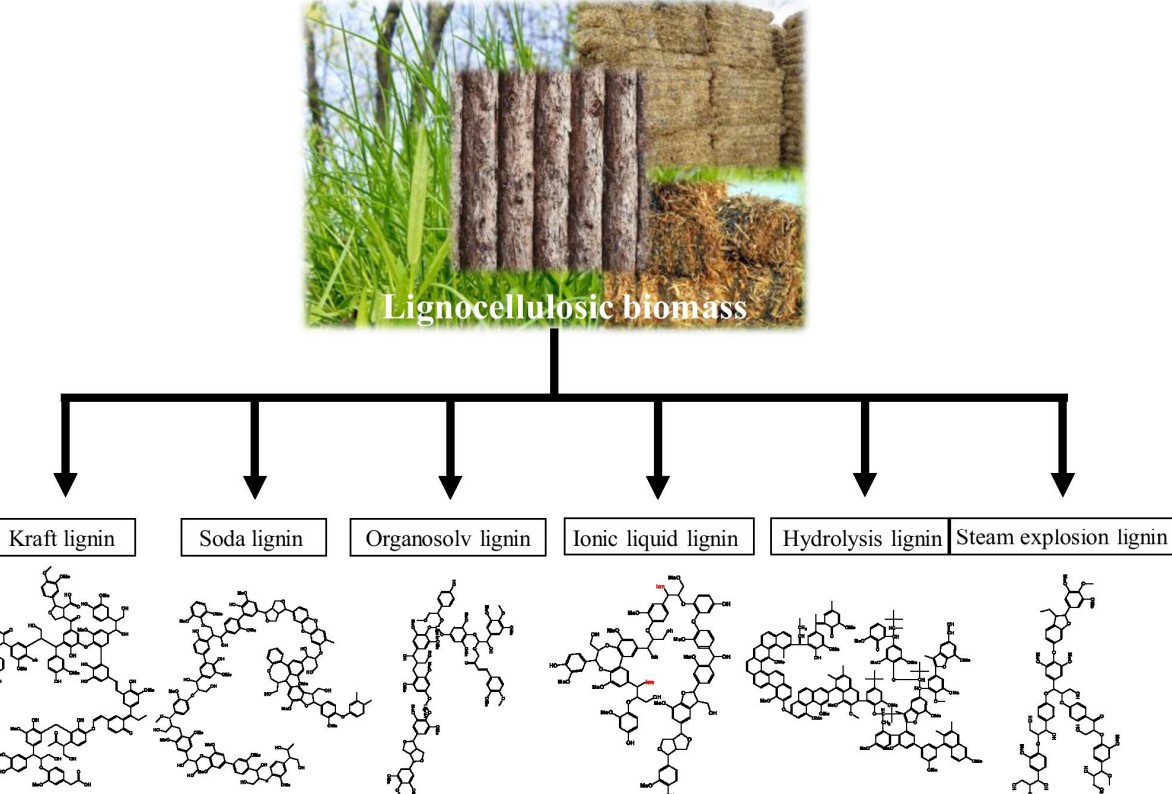

**Figure 3.** Different structures of lignin extract from lignocellulosic biomass with various processes [24].

**Table 1.** Several types of lignin and their extraction techniques.

| Types of Lignin | Extraction Conditions | | Features |
|---|---|---|---|
| | T (°C) | Extracting Reagents | |
| Kraft lignin (KL) | 150–180 | $H_2O$, NaOH, and $Na_2S$ | A highly condensed structure with low purity and contained –HS group (containing ~2–4 wt% sulfur) |
| Sulfite lignin | 120–180 | $H_2O$ and sulfites | A highly condensed structures with low purity and contained –$SO_3$ group (5–9 wt% sulfur) |
| Soda lignin (SL) | 160–170 | NaOH, $H_2O$, and anthraquinone | Sulfur-free with low purity |
| Organsolv lignin (OL) | 180–210 | alcohols), high-boiling-point solvents (glycols and glycerol), organic acids, ketones, and others | Sulfur-free with low purity |
| Alkaline lignin (AL) | 45–170 | $H_2O$, ammonia, NaOH, and $Ca(OH)_2$ | Low condensed structures |
| Enzymatic lignin (EL) | 30–60 | $H_2O$, hydrogen peroxide, cellulase, hemicellulase, additional pretreatment including reductive and oxidative, dilute acid, ammonia fiber explosion, and steam explosion | Less condensed structures with low purity |
| Dilute acid hydrolysis lignin | 120–350 | $H_2SO_4$, $H_2O$, HCl, HF, and $H_3PO_4$ | Less condensed structure with partial preserved β-O-4 linkages |
| Klason lignin | 25–120 | $H_2O$ and $H_2SO_4$ | Highly depolymerized oligomers with a condensed structure |

To gain a better understanding of the deep reaction mechanism of LDP, researchers simplified the complexity of the lignin structure to lignin model compounds with lower molecular weight. Researchers have studied lignin model compounds with a pure type of linkage, which could contribute to simplifying the mechanistic precise pathway of bond cleavages and catalytic performance [21]. Lignin model compounds usually possess the functional groups or linkages which are contained in native lignin and are usually divided into two types: monomers and dimers [28–30]. Monomer model compounds, which usually contain functional groups, are generally applied to investigate the chemical behavior of depolymerized lignin fragments. For instance, the chemical behaviors and reaction paths of methoxy, hydroxy, and alkyl groups in lignin macromolecules have been keenly evaluated by monomeric model compounds. Dimers usually contain the most typical lignin linkages, such as the β-O-4 and α-O-4 linkage model compounds. The main purpose of the studies with dimers is to assess the chemical behavior and reactivity of a particular lignin linkage, especially the bond cleavage in LDP [31].

## 3. Solvents

The choice of a solvent is a critical factor in lignin valorization, as the reaction medium is an important parameter in determining the conversion rate and the selectivity of products. These common solvents applied in LDP include water, organic solvents (alcohols, phenols, and aprotic solvents) and some co-solvent systems.

### 3.1. Aqueous Solution System

Compared with other organic solvents, water as a solvent is highly desirable due to its low cost, no toxicity, and easy recovery. Some researchers have already used water as a reaction medium for the catalytic depolymerization of technical lignin. Mushtaq and Kim compared the reaction efficiency in water with alcohol (ethanol, methanol, and isopropanol) and n-hexane under the assistance of a ZnO-Co/N-CNT catalyst [32]. Additionally, in the water medium, they obtained a 12.1 wt% monomer yield, while the yield of gas is 5.2 wt% and the yield of solid residues is 5 wt%. The monomer yield was much higher than those produced in other solvents.

When considering water as a solvent, base played an important role in this system to increase the equilibrium rate of the enolization reaction. The enolization reaction is important for the cleavage of β-O-4 and β-1 ketone under room temperature [6]. Hu et al. successfully realized the selective cleavage of C–C bonds by Cu and a base to produce

aromatic acids and phenols in water at only 30 °C. They also conclude that a base-mediated $C_\beta$–H bond cleavage is the rate-determining step for $C_\alpha$–$C_\beta$ bond cleavage.

Researchers proposed that lignin contains hydroxyl groups in its side chain, which could provide the hydrogen source for the hydrogenolysis of C–O bonds instead of other organic solvents [33]. According to this concept, Liu et al. proposed the use of Rh terpyridine complexes with outstanding hydrophobic properties as an efficient catalyst to break down the C–O bonds in water with less base [34]. It was found that this catalytic system can successfully be used to convert lignin model compounds into ketones and phenols under mild conditions.

### 3.2. Alcohols Solvent System

Alcohols, which can readily dissolve lignin, are widely used as solvents in LD, maintaining high levels of conversion without producing excessive amounts of tar and char residues [35]. In general, methanol, ethanol, and isopropanol are widely used alcohols. Alcohol conversions occur during the reaction, and the intermediates and products generated from alcohol conversions are also involved in the LD reaction [36]. In metal-catalyzed reactions, alcohol adsorbs and dissociates into active atomic hydrogen and alkoxyl groups, and then the active atomic hydrogen transforms into a substrate molecule. In those processes, alcohols could be regarded as solvents, as well as potential hydrogen donors.

Several articles have studied LD reactions in alcohol without catalysts. Ford et al. first investigated organosolv poplar lignin degraded in ethanol/isopropanol mixtures to produce phenolic lignin monomers without metal catalysts and hydrogen [37]. The yield of monomer phenols could reach 33%, while the mixed ethanol/isopropanol solvent showed synergistic effects to promote the transfer of hydrogen during the lignin hydrogenation process. Recently, technical lignin depolymerization was achieved with a 20.62 C% yield of aromatics in ethanol without catalyst and external $H_2$ [38]. Ethanol had significant solubility and H-transfer ability, which resulted in an exceptional hydrogenolysis efficiency of the β-O-4 lignin linkages.

Li et al. examined the catalytic reaction activity of LD under alcohol (methanol and ethanol) and water [39]. In methanol and ethanol, the bio-oil yields were 68.6 wt% and 71.8 wt%, respectively, which were much higher compared with the water solvent (34.6 wt%) with the same catalyst. Alcohol has a lower dielectric constant, a lower critical temperature, and a lower pressure, which results in the degradation of lignin occurring more rapidly. Furthermore, synergistic effects have been found during LD between both alcohol solvents and metal catalysts, since alcohol solvents significantly enhanced the effects of the catalyst on lignin liquefaction. In addition, alcohol also acts as a hydrogen source to provoke LD on the catalyst surface.

Under near-supercritical or supercritical conditions (pressure > 6.2 MPa and temperature > 240 °C), alcohol solvents such as ethanol are more active and can provide abundant ions or free radicals [40]. Supercritical ethanol is widely applied in catalytic processes, because it can be regarded as not only a solvent but also a hydrogen donor and capping agent [41]. In LDP, it can also help to suppress the repolymerization of phenolic intermediates, which could stabilize lignin fragments; as a result, the balance has shifted towards monomeric product production [42,43]. Therefore, supercritical ethanol is a highly potential solvent in lignin hydrogenolysis towards aromatic monomers. In supercritical ethanol, Li et al. achieved a 86 wt% selectivity of guaiacols and the conversion rate of KL was 48 wt% with $Fe_3C$/C catalysts [44]. The possible role of supercritical ethanol solvent was discussed as follows. Active metal Fe of the catalysts can activate ethanol to produce aldehydes intermediates and hydrogenate to generate aliphatic compounds. The activation of ethanol also resulted in the formation of hydrogen radicals and alkyl radicals in situ. Hydrogen radicals provided a hydrogen source for the hydrogenation and hydrogenolysis of the LD system, while alkyl radicals can act as electrophilic agents to induce the alkylation reaction of LD intermediates. Supercritical methanol was used as a solvent for LD by Kong's team. In their system, a $Cu/CuMgAlO_x$ catalyst acted as the active specie for

methanol activation [45]. Methanol simultaneously plays the role of solvent, hydrogen donor, and reactant under supercritical conditions. The C-C coupling between methanol and monomeric products during lignin hydroprocessing contributes to the methylation of aromatic rings.

Due to its capacity to provide hydrogen sources and high solubility of lignin, isopropanol is one of the most suitable options when compared with other organic solvents [46]. Gao et al. revealed $H_2$ could be used as the primary hydrogen donor during the reaction process, while isopropanol served as both the reaction medium and the secondary hydrogen donor, which could reduce the requirement for $H_2$ [47]. Fu et al. found that isopropanol provides higher selectivity to 2-hydroxy-4,6-dimethoxyacetophenone, followed by 2-methoxy phenol and 2,6-dimethoxy phenol [48]. According to other studies, the hydrogen supply capability of isopropanol facilitates the hydrodeoxygenation of phenol monomers derived from lignin [49]. Under higher temperatures, isopropanol exhibited better hydrogen transfer properties and lignin dissolving capacity, which further promoted the hydrogenolysis of lignin [50]. As a result, the reaction temperature for LD reactions with isopropanol usually exceeds 236.0 °C. Table 2 summarized recent researches on the catalytic depolymerization of lignin involving alcohols.

**Table 2.** Summary of catalytic depolymerization of lignin involving alcohols.

| Feedstock | Catalyst | Solvent | Conditions | | | Results | Ref. |
|---|---|---|---|---|---|---|---|
| | | | T (°C) | t (h) | Gas | | |
| KL | Mo(OCH$_2$CH$_3$)$_x$/NaCl | supercritical ethanol | 300 | 6 | high-purity nitrogen for six times | C$_6$ alcohols, C$_8$–C$_{10}$ esters, benzyl alcohols, and arenes yield: 30.3 wt% | [51] |
| OL from pine sawdust | Ru/C–MgO–ZrO$_2$ | supercritical ethanol | 350 | 4 | 30 bar H$_2$ | phenolic monomer yield: 31.44 wt% bio-oil yield: 76.2 wt% | [52] |
| KL | Fe-Fe$_3$C/C | supercritical ethanol | 290 | 5 | 0.5 MPa N$_2$ | guaiacol and 4-alkyl guaiacols yield: 41.2% | [44] |
| Alkali lignin (AL) | Cu@MIL-101(Cr) | supercritical ethanol | 250 | 6 | 2.5 MPa H$_2$ | aromatic monomers yield: 38.5% | [42] |
| Lignin-rich corncob residue | NiMo/Al | supercritical ethanol | 320 | 7.5 | 2.76 bar H$_2$ | aromatic monomers yield: 25.5 wt% | [53] |
| Stubborn lignin | Cu/CuMgAlO$_x$ | supercritical methanol | 300 | 4 | 5 times with pure Ar | monomeric products yield: 37.76 C% dimeric products yield: 57.97 C% | [45] |
| OL | MCM-41 supported phosphotungstic acid | isopropanol/water solvent | 310 | 6 | / | bio-oil, liquid fuels yield: 86.89 wt% lignin conversion: 95.52 wt% | [49] |
| KL | SiW$_{12}$ and Pd/C | isopropanol | 190 | 5 | / | phenolic monomers yield: 42.6% KL conversion: 95.0% | [54] |
| OL | ReO$_x$/AC | isopropanol | 200 | 3 | 100 psi N$_2$ | aromatic oils yield: 50.2–54.0% 4-propylsyringol yield: 6.5% | [55] |
| KL | Ni–Cu/H-Beta | isopropanol | 330 | 3 | N$_2$ | bio-oil yield: 98.80 wt% monomer yield: 50.83 wt% | [7] |
| ESL | Ni$_{50}$Pd$_{50}$/SBA-15 | isopropanol | 245 | 8 | 0.5 MPa N$_2$ | monophenols yield: 8.14 wt% | [56] |

### 3.3. Alcohol with Other Solvents

Several co-solvents involving alcohol with specific properties and functions in LDP have been investigated. Co-solutions may enhance lignin conversion by the formation of synergistic effects. Several pure solvents and binary solvents were used in the directional depolymerization of lignin by Yang et al. [57]. The results suggest that the residue yields under binary solvents of methanol-1,4-dioxane and methanol-dimethoxymethane were much lower than those in methanol. Additionally, the yields of total phenolic monomer under methanol-1,4-dioxane, methanol-dimethoxymethane, ethanol-1,4-dioxane, and ethanol-dimethoxymethane were much higher than those of any pure solvent. A ratio of 15:10 (*m*/*m*) of dimethoxymethane to methanol was optimized to obtain a higher yield of phenolic monomers (24.89%).

In a binary solution consisting of ethanol and 1,4-dioxane, researchers can obtain a better solubility of lignin, since 1,4-dioxane is also an effective solvent for lignin. An optimized mix-solvent containing 20 mL 1,4-dioxane, 20 mL isopropanol, and 5 mL methanol was proposed [58]. According to analyses, methanol and isopropanol are capable here of increasing the solubility of KL and providing a source of hydrogen in the reaction system, respectively. The 1,4-dioxane is essential for the dissolution of KL, and its absence degrades the performance of the catalyst. In Gu's study, co-solvents also helped reach a

good result of a low char yield (6.6 wt%) and high bio-oil yield (63.1 wt%). Their procedure utilized 1,4-dioxane as the lignin solvent, ethanol as the reactant and hydrogen donor, and formaldehyde solution as the inhibitor [59].

With the addition of formic acid, a higher yield of monomers was obtained and a much higher hydrodeoxygenation capacity was achieved, demonstrating that formic acid and isopropanol had a synergistic effect. According to Qin, formic acid served as the primary hydrogen donor and isopropanol served as the reaction medium and secondary hydrogen donor in the system [60]. By using formic acid, isopropanol, and water as solvents for the hydrodeoxygenation reaction of corn stover lignin with Co/AC-N as a catalyst, almost complete delignification (91%) was achieved, as well as a high yield of target monomers (23.8%) [61].

*3.4. Ionic Liquid System*

Ionic liquids (ILs) are considered as environmentally friendly solvents due to their particular characteristics in physical, biological, thermal, chemical, and decomposition temperature ranges. Several functionalized ionic liquids can effectively dissolve lignocellulosic materials [62]. There is widespread use of ILs in the processing of lignocellulosic materials, such as biomass pretreatment and upgradation. Some studies have pointed out that even at room temperature, ILs exhibit potential as reagents for cleaving methyl aryl ethers and β-O-4 bonds in lignin [63,64]. ILs involved in the LD process can be used as a solvent or a catalyst for the conversion of lignin into value-added products with a low molecular weight [65].

A deep understanding of the mechanism of the cleavage of β-O-4 by Brønsted acidic IL (1-methyl-3-(propyl-3-sulfonate) imidazolium bisulfate ($[C_3SO_3Hmim][HSO_4]$)) was studied by using theory calculation. The role of ILs in LD can be concluded as three potential pathways, including (1) the dehydration of α-C-OH, (2) the dehydration of γ-C-OH, and (3) the protonation of β-O [66]. N-allylpyridinium chloride ([Apy]Cl) in LD with Pt-La$_2$O$_3$–SO$_4^{2-}$/ZrO$_2$ catalysts acted as a promising solvent for lignin dissolution, as well as an intermediate medium with catalytic properties under mild conditions [3]. Meanwhile, some guaiacol was converted into catechol with the demethylation of [Apy]Cl.

Synthesis of a single IL that is bifunctional (catalyst and solvation medium) is more advantageous [67]. In order to reduce the high cost of catalysts, Li et al. explored the oxidative depolymerization of AL in a 1-ethyl-3-methylimidazolium acetate ($[C_2C_1im]OAc$) system without additional catalyst under mild conditions [68]. Approximately 77 wt% AL was depolymerized into soluble small molecular products at 100 °C. Meanwhile, the AL conversion in recovered $[C_2C_1im]$, and OAc was only decreased slightly (~4.1%) compared with fresh ILs. Gao et al. prepared CoILs with varying molar radios of $[C_2C_1im]OAc/[B_ZC_1im]NTf_2$ ($R_{IL}$) to benefit the advantages of two types of anions [69]. The highest conversion of AL was 79.8%, with high selectivity of phenol monomers (306 mg/g) at 100 °C for 2 h with $R_{IL}$ of 5/1. They indicated that in a binary IL depolymerization system, the mixtures can be designed and established to increase the selectivity of objective products. In view of the toxic nature of various imidazolium-based ILs, ammonium-based IL may be a more environmentally friendly and economical alternative to imidazolium-based ILs [70]. The potentiality of two ammonium-based aqueous ILs: *di-isopropylethylammonium chloride* ([DIPEA][Cl]) and *di-isopropylethylammoniumbenzoate* ([DIPEA] [Bn]) for LD under mild condition was confirmed [67]. The significant reductions in the average molecular weights of alkali lignin are 84.8% and 71.1% by using [DIPEA][Cl] and [DIPEA][Bn], respectively. A summary of lignin depolymerization using ionic liquids is shown in Table 3. The listed research indicates that in lignin depolymerization, ILs play a critical role in the dissolution of lignin, the effect of solvents, and the breakdown of intersubunit lignin bonds.

**Table 3.** Summary of lignin depolymerization using ionic liquids.

| Feedstock | Catalyst | Solvent | Conditions | | | Results | Ref. |
|---|---|---|---|---|---|---|---|
| | | | T (°C) | t (h) | Gas | | |
| AL | Pt-La$_2$O$_3$/SO$_4$$^{2-}$ ZrO$_2$ | [Apy]Cl | 210 | 4 | / | phenolic compounds yield: 28.7 mol% | [65] |
| AL | polyoxometalate-ionic liquid | 73.5 wt% ([C$_4$C$_1$im]HSO$_4$) in water | 150 | 5 | 2.2 MPa O$_2$ | ketone products yield: 58.4 wt% | [71] |
| Ionosolv lignin | [C$_3$SO$_3$Hmim][HSO$_4$] in MeOH/water mixture | / | 120 | 1 | ambient pressure | aromatic product yield: 87% | [72] |
| lignin | ethyl ammonium nitrate (EAN) + prolinium tetrachloromanganate(II) [Pro]$_2$[MnCl$_4$] | / | 35 | 4 h | atmospheric pressure | vanillin yield: 18–20% | [73] |
| Dealkaline | [C$_3$SO$_3$HMIM][HSO$_4$] in MeOH/water | / | 120 | 1 | ambient pressure | vanillin yield: 5 wt% apocynin yield: 8 wt% p-cymene yield: 4.8 wt% guaiacol yield: 22.3 wt% | [74] |

## 4. Metal-Based Catalysts

Catalysts are the key factor that determines the reaction effect of LD, and their selection and design directly influence product yield, distribution, and selectivity. The most common type of catalyst is the metal catalyst, due to its diversity, designability, and outstanding catalytic effect. A variety of metals have been used in the LD process, including transition metal catalysts, noble metal catalysts, and bimetallic catalysts.

### 4.1. Ni-Based Catalysts

As a transition metal, nickel is more cost-effective compared with noble metals. Meanwhile, several studies revealed that Ni-based catalysts were effective in breaking C-O and C-C bonds in lignin molecules [75,76]. This potential metal active site has been the subject of several studies involving a variety of substrates, such as activated carbon, Al$_2$O$_3$, and zeolites. Li's team developed a low-cost Ni/MgO catalyst that selectively cleaves lignin's chemical bond [77]. Results show a conversion rate of 93.4% of lignin and a monophenol production rate of 15.0%, while the selectivity of 4-ethyl phenol was 42.3% with 20% Ni/MgO. Furthermore, hydrogenolysis of the lignin model compounds is being intensively studied under the same conditions to identify a possible reaction pathway (Figure 4). The investigations demonstrate that Ni/MgO could help the selective breakage of ester bonds in lignin and the subsequent decarboxylation reaction. Their team also prepared a La$^{3+}$-doped Ni/MgO catalyst, and the conversion rate of lignin and the yield of volatile products were 84.44% and 16.50% at 270 °C for 4 h, respectively [78]. Recently, Jiang et al. successfully prepared single-atom Ni catalysts, in which single-atom Ni sites were anchored to the CeO$_2$ nanospheres with oxygen vacancies [79]. Well-dispersed Ni sites respond to hydrogen adsorption in isopropanol and release active hydrogen, effectively enhancing the hydrodeoxygenation of aromatic compounds containing methoxy groups into alkylphenols. On the other hand, lignin's C–O bonds were also activated by the abundant oxygen vacancies in CeO$_2$.

Gao et al. fabricated various catalysts containing Ni-loaded metal phosphates (Ni/MP, M = Ti, Zr, Nb, La, or Ce), while the 15 wt% Ni/ZrP showed the highest activity for the hydrodeoxygenation of lignin-derived vanillin to produce 2-methoxy-4-methylphenol(MG) [47]. According to the results, under 220 °C and 0.5 MPa H$_2$ for 30 min in isopropanol, vanillin conversion was 97.25%, while MG yield was 88.39%. They observed that the nano-sized Ni, Lewis acid sites (LAS), and Brønsted acid sites (BAS) of Ni/ZrP contributed to high catalytic activity. The dissociation of hydrogen into active hydrogen species is catalyzed by Ni, while LAS promotes the adsorption of vanillin and isopropanol, as well as their activation. BAS promoted the dehydration reaction. Zhang also proposed that different supports could affect active sites of Ni-based catalysts and thereby catalyst activities, which is due to the original acidity of the supports and the interaction between NiO and supports [80]. Consequently, Ni/HZSM-5 provided the most efficient hydrocracking of pyrolytic lignin to

light aromatics (33.8%), due to the acidic support and appropriate interaction providing sufficient Ni and acid sites, as well as the structure of micropores.

**Figure 4.** The hydrogenolysis performance of various lignin linkages differs. The following reaction conditions were used: 0.5 g of feedstock, 0.100 g of 20% Ni/MgO catalyst, 40 mL of isopropanol, 3.0 MPa $H_2$, and 270 °C for 4 h [77].

### 4.2. Pt-Based Catalysts

It is generally accepted that Pt is an active metal in the dehydrogenation process. Using a Pt/NiAl$_2$O$_4$ catalyst without hydrogen, Li et al. obtained 4-alkylphenols from native lignin [81]. The highest yield of the 4-alkylphenols was 17.3 wt% from birch lignin, and there was no obvious deactivation of the catalyst after 600 h of use. Recently, KL was first reductively depolymerized into aromatics with the Pt/HZSM-23 catalyst by Mankar's team. Due to the mesoporous structure of Pt/HZSM-23, larger molecules of lignin were transferred more efficiently, resulting in a 65.1% yield of bio-oil [82].

### 4.3. Pd-Based Catalysts

The use of metal Pd for the valorization of lignin into aromatics has attracted a great deal of attention due to the excellent catalytic properties for C–O bond cleavage [83]. Many studies reported the outstanding activity of Pd combined with other metals or/and various supports. Lv et al. prepared carbon-species-supported Pd on the MgO surface to enhance the catalytic activity towards the hydrolysis of C–O bonds [84]. They achieved 24.6 wt% of aromatics in pine depolymerization, while the selectivity of 2-methoxy-4-(propenyl) phenols was 77.2%. In order to identify the roles that metals play in LD, Karnitski et al. studied LD by bifunctional metal Pd–acid catalysts in aqueous methanol [85]. They found that efficient LDP required the presence of Pd to initiate a reaction, and catalysts with bulk Pd atoms were much more active than those with highly dispersed ones. The synergistic effect of Pd and NbOPO$_4$ facilitated the hydrogenolysis of the ester bonds in lignin-derived

oligomers, affording a high yield of 4-ethyl guaiacols [86]. A mechanistic study showed that by forming the Nb-O bond, $NbOPO_4$ stimulated the degradation of C-O bonds, whereas Pd-activated fragments contained aldehydes, carboxylic acids, and alcohols through the Pd-O interaction, thus accelerating the degradation of C-O bonds.

### 4.4. Ru-Based Catalysts

Ru is a cost-effective metal that has moderate catalytic activity but high selectivity to aromatic monomers during the depolymerization of lignin. A high-performance Ru@N-doped carbon catalyst was synthesized by Li et al., exhibiting a well-wrinkled, defect-rich, and mesoporous carbon structure and highly dispersed Ru nanoparticles [87]. As a result of the hydrogenolysis of lignin in an aqueous ethanol solvent under 300 °C and 1.0 MPa pressure of hydrogen, this catalyst demonstrated a high yield of aromatic monomers of 30.5%. Recently, Ding and his coworkers also explored Ru supported on N-doped (Ru@N-Char) and N, P co-doped biomass-derived chars (Ru@NP-Char) as a catalyst for the depolymerization of softwood and hardwood [88]. Totals of 57.98 wt% and 17.53 wt% of phenolic monomers were obtained from the depolymerization of poplar and pine, respectively.

Jiang et al. prepared a Ga-doped HZSM-5-supported Ru catalyst with strong interaction and achieved a high liquid yield [89]. The highly active metal Ru could control the cleavage of the C-O bond in lignin model compounds and the subsequent hydrogenation of the aromatic rings.

### 4.5. Bimetallic Catalysts

By adding another metal to a monometallic catalyst, the catalyst properties of the catalyst can be enhanced in comparison with those of a monometallic catalyst. On the other hand, from an economic perspective, the precious metal catalyst is always replaced with cheaper and more abundant nonprecious metal catalysts, such as nickel, copper, and molybdenum. Furthermore, it has been proved that combining the advantages of two metals could enhance catalytic activity by producing a synergistic catalytic effect. There are several benefits associated with bimetallic catalysts, which are summarized as follows: (i) improving the catalytic activity, (ii) improving the stability of the catalyst, and (iii) improving the selectivity of the catalyst [24]. The works on the catalytic depolymerization of lignin using bimetallic catalysts are summerized in Table 4.

**Table 4.** Summary of catalytic depolymerization of lignin using bimetallic catalysts.

| Feedstock | Catalyst | Solvent | Conditions | | | Results | Ref. |
|---|---|---|---|---|---|---|---|
| | | | T (°C) | t (h) | Gas | | |
| Lignin model compound | Ni-CeO$_2$/H-ZSM-5 | ethanol | 150 | 2 | 1 MPa H$_2$ | ethylbenzene yield: 63.4% | [90] |
| OL | La-doped Ni/MgO | isopropanol | 270 | 4 | 3.0 MPa H$_2$ | volatile products yield: 16.50% | [78] |
| Acid-extracted birch lignin | PtRe/TiO$_2$ | isopropanol | 240 | 12 | H$_2$ | monophenols yield: 18.71 wt% 4-propylsyringo yield: 7.48 wt% | [91] |
| Kraft lignin | ReMo@ zeolitic imidazolate framework nanocatalyst | 1, 4-dioxane+ methanol | 300 | 24 | H$_2$ | biofuels yield: 78% | [92] |
| OL | NiCu/C | ethanol/isopropanol | 270 | 4 | 1 MPa N$_2$ | phenolic monomers yield: 63.4 wt% | [93] |
| Birch lignin | Ni$_{50}$Pd$_{50}$/SBA-15 | isopropanol | 245 | 4 | / | monophenols yield: 37.2 wt% | [94] |
| KL | Ni-Ce/BC | glycerol/water = 1/6 | 280 | 4 | 0.5 MPa N$_2$ | lignin oil yield: 59.02% guaiacol yield: 243.94 mg/g lignin 4-alkyl guaiacols yield: 265.65 mg/g lignin | [95] |
| AL | Ni-Co/AC | ethanol | 280 | 0.25 | / | bio-oil yield: 72.0 wt% | [96] |
| OL | 5%Pt-1%Ni/HTC | ethanol/water = 45.9% (*v/v*) | 233 | 1.47 | / | 18 wt% lignin oil fraction with 72 wt% lignin tar fraction | [97] |
| AL | Ni$_x$Zn$_{1-x}$/ZrO$_2$-MgO | formic acid and isopropanol | 240 | 6 | / | bio-oil yield: 65.22 wt% alkylphenol yield: 13.22 wt% with 56.97% of selectivity | [60] |

Cheng et al. introduced Cu in NiCu/C catalysts to increase reactivity. With 5 wt% of Cu content, the highest monomer yield was 63.4 wt% without precious metal catalysts and external hydrogen [93]. With the introduction of Cu, the hydrogen donor process of

ethanol/isopropanol was promoted, and alcohol solvent consumption was minimized. Thus, the presence of Cu in catalysts could result in the cleavage of lignin linkages and the lowering of the molecular weight of bio-oil. When introducing Pd in a $Ni_{50}Pd_{50}$/SBA-15 catalyst, a yield of 37.2 wt% was obtained by Hu et al., which is 6.1 times greater than that of Ni/SBA-15 and 1.5 times greater than Pd/SBA-15 [94]. Compared with a Ni monometallic catalyst, Ni–Pd bimetallic catalysts provided more electron-rich Ni sites and exhibited higher activity, which may be attributed to its uniform distribution with relatively small size and high ability to catalyze the production of hydrogen from isopropanol in situ.

Different kinds of phenolic compounds were obtained over bimetallic Ni-based catalysts with different combinations of Fe, Zn, Co, Mo, and Cu [98]. It was found that incorporation resulted in a higher yield of lignin-derived bio-oils (60 wt%) with higher H:C ratios (1.38) and a greater proportion of phenolic compounds (from 40% to 70%). A higher selectivity for guaiacols can be observed over Ni/Zn, while a low level of eugenol was observed with the introduction of Fe active sites, but a high degree of selectivity was observed towards 2-methoxy-4-propyl-phenol (over 4.0%). Other metallic sites show low deoxygenating catalytic activity, resulting in high yields of oxygen-containing guaiacols.

An easy one-step procedure was proposed for preparing $Ni_xCo_{1-x}$/C catalysts derived from MOFs by Zhu et al.; the highest yield of monophenols was 55.2%, and the guaiacol selectivity was as high as 70.3% [99]. The outstanding performance might be due to the hydrodeoxygenation (HDO) activity enhancement of cobalt.

Lin et al. studied the catalytic activity and reaction mechanism of the lignin of Ni-$CeO_2$/H-ZSM-5 catalysts. The addition of $CeO_2$ increased the dispersion of Ni on the surface of H-ZSM-5 as well as the oxygen vacancies to increase the catalytic activity. The highest ethylbenzene selectivity was 63.4%, which means the catalyst showed a strong selective catalytic cracking for the β-O-4 chemical bonding structure to produce ethylbenzene chemicals [90]. Feng et al. also demonstrated that the active phase of $CeO_2$ introduced to Ni/H-ZSM-5 catalysts can facilitate the catalytic cleavage of the β-O-4 bond in lignin. The redox properties of active species and their surface valence states were significantly influenced by the introduction of $CeO_2$.

## 5. Supports of Catalysts

Various supports, such as activated carbon, silica, alumina, and zeolite, are used for the design of catalysts. Numerous physical and chemical properties, as well as a large surface area and topological structure, could improve dispersibility and change the physicochemical properties of catalysts, and the catalytic potential is maximized.

### 5.1. Carbon Materials

As support, carbon-based materials have outstanding properties, such as excellent electrical conductivity, a large specific surface area, and a microstructure that can be adjusted [100]. The impressive mechanical nature of carbon supports, as well as the easy accessibility of the active metal phases both promote the carbon materials to become excellent catalyst supports [101].

### 5.1.1. N-Doped Carbon

As an electron donor, nitrogen atoms could increase the reduced work function of a catalyst in comparison with pristine carbon support. Nitrogen-doped carbon-supported catalysts consist of a high density of oxygen-containing functional groups, leading to outstanding dispersion properties and excellent performance [102]. Furthermore, the increase in the electron transfer ability between the metal and support contribute to the enhancement of the binding, as well as an increase in chemically active sites. In addition, nitrogen atoms also enhance the dispersion of metal nanoparticles, as well as inhibit their aggregation. As both a foaming agent and nitrogen source, Wang et al. employed ammonium oxalate to carbon materials to increase their surface area (608.74 $m^2$/g). As a result, Ru and $WO_x$ nanoparticles on the surface were found to be more stable and dispersive [103]. Carbon

surfaces incorporated with nitrogen showed an increase in hydrophilicity and structural defects, facilitating ether bond cleavage.

Recent research revealed that N-doped carbon coordinated with metal catalysts displayed improved activity and stability, because metal atoms and N atoms are strongly interconnected electronically [104]. Li et al. prepared a single-atom Ni@N-carbon catalyst with a two-step pyrolysis process by introducing N species into C carriers. Ni sites are formed atomically dispersed, and N coordination sites are provided between Ni and N species [105]. Atomically dispersed Ni decreases the adsorption energies of Ni on the C carrier, as well as hydrogen sources' dissociation energy and transition state energy barriers. As a result, the high-Ni loading catalyst exhibited outstanding catalytic activity and durability.

### 5.1.2. Biochar

A biochar (BC) produced from biomass-based resources has several advantages over other carbon-based materials, including its low cost, accessibility, and sustainability [48]. As a support material, its surface has numerous oxygen-containing functional groups. The metal ions would be adsorbed by these functional groups because of their coordination and electrostatic properties [106,107]. Furthermore, using biochar as a reducing agent could result in a reduction in metal precursor salts into metal nanoparticles or the formation of surface defects in metal oxides [108]. Chen's group prepared 3Ni-Ce/BC and achieved a high yield of guaiacol (243.94 mg/g lignin) and 4-alkyl guaiacols (265.65 mg/g lignin) [95]. Based on their characterization results, BC promoted metallic Ni site and oxygen vacancy formation on Ni-CeO$_{2-x}$ interfaces, which could promote the adsorption and activation of C-C and C-O bonds and further depolymerized lignin fragments to form reactive intermediates.

### 5.1.3. Activated Carbon

The excellent porosity and large surface area of activated carbon (AC) promote its use as a catalytic support. AC used as support has considerable advantages in hydrogenation, cyclization, and isomerization reactions [109]. Zhang et al. found that when using Ni–Cu/AC as a bimetal catalyst for LD, no excessive hydrogenation products were found. The result is probably due to the fact that the acidity of the AC support may cause excessive hydrogenation of the benzene ring [110]. Biswas et al. activated low surface area and activity biochar into AC by KOH mixture and calcine [96]. Biochar-derived AC-supported bimetallic catalysts led to a maximum bio-oil yield of 72.0 wt%.

### 5.1.4. Carbon Nanotube

Carbon nanotubes with easy modification are hydrothermally stable and interact moderately with active metal sites, which are widely applied as superior catalyst supports. In Wu's work, various groups of functionalization on carbon nanotubes were applied as supports of Ru metal, which provided possible effects on the chemical states of Ru nanoparticles. According to the results, a high yield of 27.12% of phenolic monomers can be obtained by using CNT-NH$_2$ to form the finest Ru nanoparticles [111].

### 5.1.5. Graphene Oxide

Both industry and research have shown interest in graphene oxide (GO) due to its specific physicochemical characteristics, such as high strength, physical stability, and specific surface area, as well as high electron mobility and thermal efficiency [112]. In addition, as GO is rich in oxygen-containing functional groups, including epoxides, phenolic hydroxyl, carboxylic, and other carbonyl groups, it exhibits outstanding catalytic activity in LD [113]. Totong et al. demonstrated an improved catalytic activity of GO-supported metal oxides' bagasse lignin decomposition [113,114]. The various functional groups could absorb lignin molecules through an ether bond, followed by the formation of a hydrogen bond [115]. GO's acidic and oxidizing properties also contribute to its high catalytic activity.

The chemical reduction of GO obtained rGO with residual oxygen functional groups and electron defective sites [116]. $C_\alpha-OH$ in lignin model compounds could be oxidized into aldehydes with RuCo/rGO composites as catalysts [117]. Electron-rich rGO sheets activate RuCo to oxygen molecules, forming superoxide ions, which could absorb hydrogen [118]. In addition to interacting with aromatic substrates through π–π stacking, rGO can also enhance lignin oxidation.

### 5.2. Metal Oxides

A lot of attention has been paid to metal oxides for use as carriers for LD due to their acidities, basicity, surface areas or valences, such as $Al_2O_3$, $ZrO_2$, $WO_3$, and $CeO_2$.

$Al_2O_3$ has a specific surface area and acidic properties which facilitate the adsorption and dissociation of oxygenated groups [119,120]. With the assistance of the NiRu/$Al_2O_3$ catalyst, Yan et al. reached a high yield of monomeric phenols (38.1%), which was twice that of the reaction without the catalyst [121]. When incorporating $Al_2O_3$ together with other metal oxides (such as $Nb_2O_5$, $WO_3$, $TiO_2$, and $CeO_2$), it can be used as a support material. By modifying the physicochemical properties of the support, including active site type, surface area, and total acidity, Kong et al. increased the acidity of the support [122,123]. A series of inexpensive bimetallics were doped on $WO_3-Al_2O_3$, which contained both Lewis and Brønsted acidic sites [98]. Alumina frameworks with tungsten-enhanced surface acidity and textural properties were modified, including pore volume and surface area.

Co/$CeO_2$ had an outstanding performance in the selective cleavage of β-O-4 linkages under mild conditions without hydrogen [116]. $C_\alpha$-OH was converted into acarbonyl group, leading to the generation of acetosyringone at a rate of 78%. Meanwhile, the catalyst had good stability and catalytic activity after four cycles, which means it showed no obvious deactivation. $Nb_2O_5$ support also could select the conversion of C-OH bonds in lignin because of the excellent oxophilicity [124]. Thus, Kong et al. prepared a recyclable Ni-Re/$Nb_2O_5$ for a selective and efficient process for depolymerizing KL in an ethanol solvent. The results obtain the highest oil yield of 96.70 wt%, with less char formation [125].

### 5.3. Zeolite

Zeolite is an ideal catalyst support for the enhancement of LD, resulting from its larger surface area, well-organized pores, and degree of acidity. Kong et al. prepared Ni–Cu bifunctional catalysts on different kinds of zeolites, including Ni–Cu/H-Beta, Ni–Cu/HZSM-5, Ni–Cu/MAS-7, Ni–Cu/MCM-41, and Ni–Cu/SAPO-11 to study the influence of pore structure and acidity on monomer yield, char yield, and product distribution. All the zeolite supports had relative activity in LDP, while a KL conversion of over 90% was observed for Ni–Cu catalysts supported by H-Beta, MAS-7, and MCM-41 [7]. Therefore, the large mesoporous surface area of the support contributed to the dispersion of the catalyst and the conversion of lignin-derived oxygenates. The highest oil yield was found for the Ni–Cu/H-Beta (98.80 wt%). Xu et al. also compared the performance of MCM-41, β-zeolite, HZSM-5, HY, ZnO/HZSM-5, and ZnO/HY in wheat straw lignin valorization. The pore structure and acidity of the ZnO/HZSM-5 catalyst made it optimal for producing aromatic amines [126].

HZSM-5, a well-defined crystalline microporous (<2 nm) aluminosilicate material, is widely used in the production of aromatics due to its specific porous structure, controllable acidity, and selectivity towards particular products [127]. Microporous HZSM-5 zeolite has been considered as one of the most efficient catalysts for lignin catalytic reactions benefiting from appropriate acidic site distribution and shape selectivity [128]. With strong deoxygenation capability, HZSM-5 is conducive to the hydrodeoxygenation of lignin and improves the octane value of depolymerization products [129]. Zhu et al. first investigated the influence of the hierarchical mesoporous physical property of HZSM-5 on the LD. Hierarchical HM5 zeolites were prepared by alkali treatments with different NaOH concentrations. The hierarchical HZSM-5 zeolite catalyst showed an increasing catalysis effect with high stability [130]. By impregnating highly dispersed Fe–Pd bimetallic

catalysts onto HZSM-5, Zeng et al. achieved a 27.92% aromatic monomer yield [131]. Liu et al. prepared Ni/HZSM-5 catalysts with different Ni loadings. The Ni/HZSM-5 catalysts were not only responsible for the cleavage of β-O-4 linkages but also for hydrogenated unsaturated intermediates [132].

The application of the MCM-41 zeolite as a support could enhance the activity of the prepared catalysts owing to its ability to control both electron- and charge-transfer processes and stabilize the charge-transfer state of transient species [133–135]. Lu et al. prepared a NiMo/Al-MCM-41 catalyst and obtained a bio-oil yield of 61.6 wt% with high selectivity to alkyl guaiacols [136]. The synergic action of acidic support and metal active sites enhanced the cleavage of ether linkages and increased the yield of phenolic monomers.

*5.4. Silicate*

Sepiolite (SEP) is a kind of fibrous hydrated magnesium silicate which belongs to the sheet silicate (phyllosilicate) group of silicates [137]. SEP's structure consists of continuous tetrahedral sheets and discontinuous octahedral sheets, which have numerous channels to accommodate water and organic molecules [138,139]. Due to their fibrous crystalline structure, high thermal stability, and lower price, SEPs have been attracting a lot attention in catalytic fields [140]. Chen's team intensively studied the performance of SEP-supported catalysts in LD, especially Mo-based catalysts [141]. The highest soluble fraction yield of petroleum ether and the yield of liquid product were 47.6% and 63.5%, respectively. The reactive performance under a series of calcination temperatures of Mo/SEP catalyst has also been researched. The results show that calcination temperature could influence the transformation of Mo species from $(NH_4)_6Mo_7O_2 \cdot 4H_2O$ to $MoO_3$ and subsequently to $Al_2(MoO_4)_3$, while the temperature varies from 200 °C to 600 °C [142]. The transformation could affect the amount of active $Mo^{5+}$ species and the distribution of B/L acidic sites on the Mo/SEP surface. Therefore, a yield of 80.29% of lignin oil and a yield of 65.74% of petroleum ether-soluble products were obtained with Mo/SEP calcined at 400 °C. The performance of the W–Ni–Mo/SEP trimetallic catalyst has been further studied by the same team [143]. Compared with Mo/SEP, W–Ni–Mo/SEP with unique cyclic stability could promote the production of fuel and phenol monomers and the selectivity to guaiacol and methoxyphenol. Metallic oxide phases were transformed by doping W and Ni metals; as a result, the metal–support interaction was enhanced, oxygen vacancies were increased, and acid sites were decreased. SEP-supported bimetallic Mo–Mn catalysts with different Mn/Mo ratios were also prepared and applied to LD. In 3Mn1Mo/SEP, the appropriate interaction among SEP, Mn, and Mo species reduced the particle sizes of metal oxides and rendered the crystalline phase structure, which reached 45.7% of lignin oil yield and 308.9 mg/g lignin of guaiacols yield [144].

## 6. Summary and Prospect

Lignin is considered a promising source of renewable raw materials which can produce bio-oil and high-value chemicals.

In this review, lignin valorization methods and strategies with different solvents and catalysts were overviewed. A great deal of research has been conducted on the high-efficiency conversion of lignin into low-molecular-weight compounds. However, there are still many urgent problems that need to be paid more attention to. First, although a large amount of research has focused on LDP, clear and comprehensive reaction mechanisms have not been verified yet, especially for native lignin. Second, lignin has been converted catalytically and a high yield of products has been obtained, but the selective conversion toward the specified products is also a big challenge. Meanwhile, the purification of conversion products is also an important factor for the effective utilization of lignin. Thirdly, another problem for LD is that oxidation or carbon deposits could cause catalyst deactivation. Last, at present, the conversion and degradation of lignin are mostly in the laboratory, and there is still a long way to realizing industrialized production.

In summary, there are still many new opportunities that exist for developing a high-efficiency system for LD. According to these problems, future investigations could focus on the following aspects:

(1) To develop one-pot processes to obtain high-value chemicals and biofuels directly from lignin without pretreatment.

(2) To improve catalyst activity for cleaving C-C bonds and further improve reaction product selectivity and optimize product distribution.

(3) To deeply study the breaking process of the internal chemical bonds and the mechanism of depolymerization reaction, especially for native lignin. Meanwhile, further research is needed on the strategies which could suppress the repolymerization of lignin.

(4) To investigate cheaper catalysts and economic systems to easily separate and regenerate the catalyst after specified cycles of reactions.

**Author Contributions:** Y.W. drafted the original paper and conducted the literature review; W.L. provided some literature research and assisted in the writing of the review; L.W. and Q.H. provided some articles and studying materials; Z.M. and X.L. assisted with literature research and supervised the project. All authors have read and agreed to the published version of the manuscript.

**Funding:** This research was funded by Inner Mongolia Autonomous Region Scientific and Technological Achievements Transformation Project(2019CG105) and the Open Research Subject Foundation of Xinjiang Biomass Solid Waste Resources Technology and Engineering Center (Kashi University) (KSUGCZX202203).

**Institutional Review Board Statement:** Not applicable.

**Informed Consent Statement:** Not applicable.

**Data Availability Statement:** Data available on request from the authors.

**Conflicts of Interest:** The authors declare no conflict of interest.

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
