# Peer review of "A Review on Catalytic Depolymerization of Lignin towards High-Value Chemicals: Solvent and Catalyst"

_fermentation, doi:10.3390/fermentation9040386_

Round 1

Reviewer 1 Report

Thank you for the opportunity to review the article entitled “A Review on Catalytic Depolymerization of Lignin Towards High‑value Chemicals: Solvent and Catalyst” which deals with a relevant and important topic.

I appreciate the experimental work in the paper and I feel that this research throws new light that deserves publication in a journal devoted to the latest finding in academia and industries research on lignin chemistry.

The article made a positive impression on me. It is clear that the authors have done a great job. I miss the review of the different types of lignin, mainly of methods for isolation and determination of lignin from wood, pulp, black liqueur, agricultural residues, etc.

Also the authors need to strengthen the link with the circular economy, bioeconomy to provide a rich resource for future fundamental and applied research. Based on these additions, the article can be accepted for publication.

Author Response

On behalf of my co-authors, we thank you very much for your positive respond to our work and valuable suggestions to our manuscript, the main corrections in the paper and the responds to the comments are as following:

Point 1: I miss the review of the different types of lignin, mainly of methods for isolation and determination of lignin from wood, pulp, black liqueur, agricultural residues, etc.

Response 1: As the reviewer suggested, we’ve performed a table summarized several types of lignin and their extraction techniques in our revised manuscript (Table 1). As we mainly focused on the catalytic process of lignin, so we don't address too much introduction about technical lignin.

Point 2: Also the authors need to strengthen the link with the circular economy, bioeconomy to provide a rich resource for future fundamental and applied research

Response 2: Thanks for your constructive suggestion, we’ve added some market analysis on lignin and its high value products to provide some economic evidence about the significance of lignin utilization (Line 90-99 and Figure2).

Reviewer 2 Report

I hope that the following information will be helpful in deciding the fate of the presented article : fermentation - 2333879

This publication deals with the very important issue of valorization of lignin due to its complex structure. Numerous works have been presented in a thoughtful, understandable way to find a promising and efficient way to valorize lignin into high-value chemicals. Studies involving both metal catalysts with different supports as well as the solvents themselves and their effect on the efficiency of the lignin depolymerization process have been extensively presented. I believe that the presented material is very valuable, as it presents important achievements related to developments in the field of catalysts, thereby providing important explanations and models according to various theories.

In my opinion, the article " A Review on Catalytic Depolymerization of Lignin Towards High‑value Chemicals: Solvent and Catalyst" has a high scientific value and can be accepted for publication after making the necessary corrections.

 Nevertheless, it is advisable that the work be read again, for example, by a native speaker, and that the necessary linguistic corrections be made.

Author Response

On behalf of my co-authors, we thank you very much for your positive respond to our work and valuable suggestion to our manuscript, the responds to the comments are as following:

 Point 1: It is advisable that the work be read again, for example, by a native speaker, and that the necessary linguistic corrections be made.

Response 1: We found there are some obvious errors in the English expression of this work. We are very sorry for our negligence of this detail, we’ve revised these sentences in our revised manuscript.